# Reconstruct and Match: Out-of-Distribution Robustness via Topological Homogeneity

**Chaoqi Chen**[1]    **Luyao Tang**[2]    **Hui Huang**[1][*]

[1]College of Computer Science and Software Engineering, Shenzhen University
[2]School of Informatics, Xiamen University
cqchen1994@gmail.com, lytang@stu.xmu.edu.cn, hhzhiyan@gmail.com

## Abstract

Since deep learning models are usually deployed in non-stationary environments, it is imperative to improve their robustness to out-of-distribution (OOD) data. A common approach to mitigate distribution shift is to regularize internal representations or predictors learned from in-distribution (ID) data to be domain invariant. Past studies have primarily learned pairwise invariances, ignoring the intrinsic structure and high-order dependencies of the data. Unlike machines, humans recognize objects by first dividing them into major components and then identifying the topological relation of these components. Motivated by this, we propose Reconstruct and Match (REMA), a general learning framework for object recognition tasks to endow deep models with the capability of capturing the topological homogeneity of objects without human prior knowledge or fine-grained annotations. To identify major components from objects, REMA introduces a selective slot-based reconstruction module to dynamically map dense pixels into a sparse and discrete set of slot vectors in an unsupervised manner. Then, to model high-order dependencies among these components, we propose a hypergraph-based relational reasoning module that models the intricate relations of nodes (slots) with structural constraints. Experiments on standard benchmarks show that REMA outperforms state-of-the-art methods in OOD generalization and test-time adaptation settings.

## 1 Introduction

Distribution shift has emerged as a major challenge for the success of machine learning systems [34, 25]. Although deep learning models are believed to generalize well to in-distribution (ID) data, a well-trained model deployed in the open world often encounters out-of-distribution (OOD) data whose contexts may differ from the training distribution, resulting in dramatic performance degeneration and raising concerns about model safety in many high-staking applications, such as autonomous driving and medical diagnosis. This gives rise to the importance of OOD generalization [92, 76], which aims to build a robust learning machine that can perform well in unseen test environments.

Regarding OOD generalization, a central theme is how to learn general features from training data that can be extrapolated to test distributions. Following this idea, a plethora of OOD generalization methods have been proposed over the past few years, including domain alignment [42], latent feature disentanglement [61, 48, 85], meta-learning [40, 41], invariant risk minimization [2, 1, 96], and augmentation-guided invariant predictor [74, 83, 95], to name a few. On the other hand, given the natural adaptivity gap between training and test distributions [15], recent works [29, 30, 81, 11, 8] attempt to further enhance the source-trained model through test-time adaptation [46], which leverages unlabeled target samples to update the model in an online manner. These two series of studies address distribution shifts during training and inference and have been proven to work synergistically [8].

---

[*]Corresponding author

38th Conference on Neural Information Processing Systems (NeurIPS 2024).

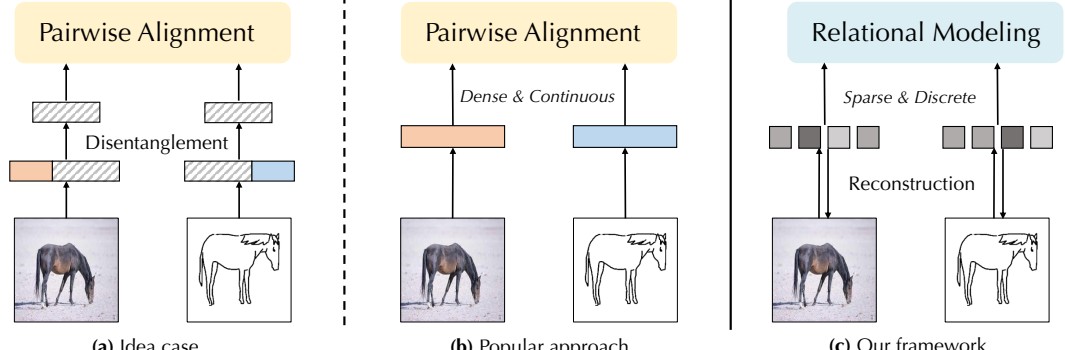

Figure 1: **Motivation of the proposed REMA. (a)** Ideally, by partitioning the latent factors into common and specific parts, aligning the common factors directly enables perfect matching of two different distributions. **(b)** Since the latent features learned by deep networks are typically dense and continuous, most well-performing methods seek direct alignment without exploring the inherent structures. **(c)** Our REMA introduces a sparse and discrete element – slot [51] – to serve as a bridge for both reconstruction and relational modeling, avoiding explicit disentanglement in the latent space.

Albeit their general efficacy for various tasks, these prior efforts largely overlook the topological structure of the image data. As a consequence, existing OOD generalization methods highly rely on specialized regularization objectives and are significantly less interpretable than human vision systems. Unlike machines, human object recognition [4] typically involves initially decomposing objects into several major components (*e.g.,* keypoints), followed by identifying the structural relationships among these components, and finally making predictions by comprehensively considering the main components and their inherent relations. To this end, a critical question remains open in the field:

*How to devise a unified framework that imitates the human vision process for OOD generalization?*

In this work, we believe that OOD generalization requires consideration of two key aspects: 1) how to represent the structure of data, and 2) how to model the relationships between different entities.

Grounded on these insights, we propose Reconstruct and Match (REMA), a general framework to endow deep models with the capability of capturing the *topological homogeneity* of objects without using human prior knowledge or fine-grained attribute annotations. Figure 1 illustrates the motivation. To identify major components from objects, REMA introduces a slot-based reconstruction module to map dense pixels into a sparse set of slot vectors in an unsupervised manner. This module encourages the deep model to reduce unwanted redundancy and preserve those predominant parts (the words "component" and "part" are used interchangeably). Then, to model and reason the high-order dependencies among these components, we propose a hypergraph-based matching module that discovers the relations of slots with structural constraints, *i.e.,* topological homogeneity of the object. The main contributions of this paper are summarized as follows:

- We propose REMA, a novel OOD generalization framework to mitigate distribution shifts in deep learning models by imitating the human visual recognition process.
- We introduce a self-supervised reconstruction process to identify informative parts from objects without additional supervision, and a hypergraph matching module to reason the high-order part-based object relationships and interactions across domains.
- Experiments on six widely used benchmarks demonstrate that REMA outperforms state-of-the-art methods in OOD generalization and test-time adaptation settings.

## 2   Preliminaries

**Problem Setup.** Assume that $\mathcal{X}$ is the input space, $\mathcal{Z}$ is the latent space, and $\mathcal{Y}$ is the output space. The predictor $f = h \circ g$ is comprised of a featurizer $g : \mathcal{X} \mapsto \mathcal{Z}$ that learns to extract embedding features, and a classifier $h : \mathcal{Z} \mapsto \mathcal{Y}$ that makes predictions based on the extracted features. The goal

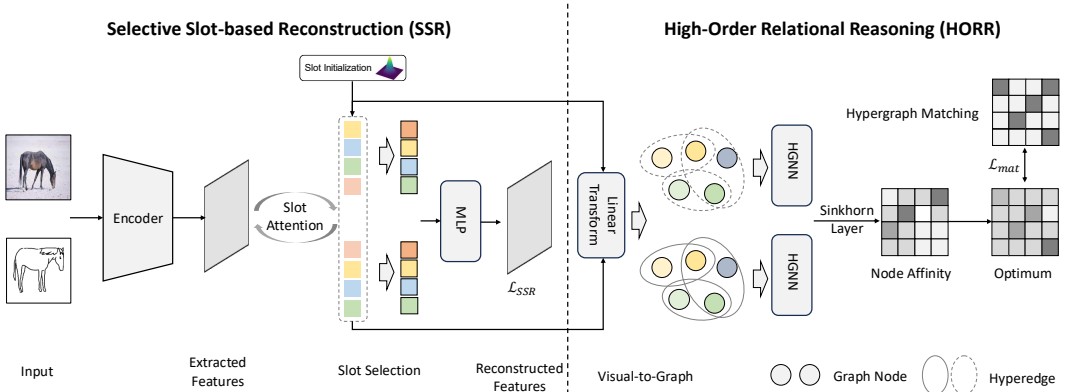

Figure 2: Overview of the proposed REMA, which consists of two key modules, *i.e.,* SSR and HORR. (1) **Abstraction:** Slot-based reconstruction to discover the main components from the data by binding objects with a set of discrete vectors; (2) **Reasoning:** Introduce high-order relational inductive bias (*i.e.,* topological homogeneity) to the network via the process of hypergraph construction, learning, and matching. HGNN means hypergraph neural networks.

of OOD generalization is to find a predictor $f : \mathcal{X} \mapsto \mathcal{Y}$ that generalizes well to all unseen target domains. In deep neural networks, empirical risk minimization (ERM) [72] is capable of learning highly predictive features, making it the simplest baseline for this problem.

**ERM Objective.** We train the model by minimizing the empirical cross-entropy loss function:

$$\arg \min_{\theta \in \Theta} \mathbb{E}_{(x,y) \sim P^{tr}} [\ell(f_\theta(x), y)]. \tag{1}$$

where $P^{tr}$ is the training distribution, $\Theta$ denotes the parameter space, and $\ell$ is the loss function. The trained model will be evaluated on a test set from a test distribution $P^{ts}$, where $P^{tr} \neq P^{ts}$.

## 3 Proposed Method

In this section, we provide a detailed description of our Reconstruct and Match (REMA) framework. As shown in Figure 2, REMA consists of two key modules tackling the main component discovery (Section 3.1) and high-order relational modeling and reasoning (Section 3.2).

### 3.1 On Discovering Main Components

For OOD generalization/adaptation, a long-standing issue is how to distinguish between transferable (domain-agnostic) and non-transferable (domain-related) information in the data. An ideal solution is to disentangle latent representations in an unsupervised manner (see Fig. 1(a)), which has been proven to be prohibitively difficult and even infeasible [50]. Therefore, most prior works (see Fig. 1(b)) aim to learn domain-invariant representations through elaborate feature alignment modules, such as adversarial training [21] and pseudo-labeling [37], without exploring the intrinsic composition of the features themselves. Despite their general efficacy, these methods may introduce a significant amount of redundant and noisy information during the alignment process. For humans, when identifying an object, we tend to first look for its main components, which can be regarded as a process of information compression from a learning perspective. Inspired by this, we aim to mimic the very first step of the human recognition process by identifying the main components from object images without human prior knowledge or fine-grained annotations.

To be specific, we introduce a self-supervised reconstruction module based on Slot Attention [51], termed *Selective Slot-based Reconstruction (SSR)*, to assist deep models in learning a set of part-based representations for sparsely characterizing a target object. Formally, given an input $x$, $g$ extracts a set of feature embeddings $\mathbf{z} \in \mathbb{R}^{N \times d_z}$, where $N$ is the number of embedding and $d_z$ denotes embedding dimension. The Slot Attention module will first take a set of slots $\mathbf{s} \in \mathbb{R}^{K \times d_s}$ ($K$ is the number of slots) and the feature map $\mathbf{z}$, then project them to dimension $d_s$ by a linear transformation $f_k$ for slots

and $f_q$, $f_v$ for $\mathbf{z}$. The Slot Attention will be trained as follows,

$$\text{update}(\boldsymbol{A}, \mathbf{v}) = \boldsymbol{A}^T \mathbf{v}, A_{ij} = \frac{\text{attn}(\mathbf{q}, \mathbf{k})_{ij}}{\sum_{l=1}^{K} \text{attn}(\mathbf{q}, \mathbf{k})_{lj}}, \text{attn}(\mathbf{q}, \mathbf{k}) = \frac{e^{M_{ij}}}{\sum_{l=1}^{N} e^{M_{il}}}, \boldsymbol{M} = \frac{\mathbf{kq}^T}{\sqrt{d_s}}, \quad (2)$$

where $\mathbf{q} = f_q(\mathbf{z}) \in \mathbb{R}^{K \times d_s}$, $\mathbf{k} = f_k(\mathbf{z}) \in \mathbb{R}^{N \times d_s}$, and $\mathbf{v} = f_v(\mathbf{z}) \in \mathbb{R}^{N \times d_s}$ denote the query, key and value vectors respectively. $\boldsymbol{A} \in \mathbb{R}^{N \times K}$ stands for the attention matrix. Unlike self-attention, the queries in slot attention are a function of the slots $\mathbf{s} \sim \mathcal{N}(\mathbf{s}; \boldsymbol{\mu}, \boldsymbol{\sigma}) \in \mathbb{R}^{K \times d_s}$, and will be updated iteratively over $T$ iterations. The slots are initialized by random sampling from a standard Gaussian distribution. Queries at iteration $t$ are represented by $\hat{\mathbf{q}}^t = f_q(\mathbf{s}^t)$, and the slot updating process is: $\mathbf{s}^{t+1} := \text{update}(\text{attn}(\hat{\mathbf{q}}^t, \mathbf{k}), \mathbf{v})$. After each iteration, a Gated Recurrent Unit (GRU) is employed on the slot representations $\mathbf{s}^{t+1}$ to update their states, integrating new information while retaining relevant context from previous iterations.

Since the number of slots $K$ is manually predefined and fixed during training, attention networks may learn redundant or even incorrect associations, thus affecting the understanding of objects and scenes. Therefore, our SSR introduces a selective slot attention mechanism that dynamically adjusts the importance of slots for reconstruction. To quantitatively represent which slots are more important for reconstruction, we first introduce an importance score $\rho$ for each slot. For each slot $\mathbf{s}_i$, we use a lightweight neural network $h_\theta$ to predict an initial importance score: $\rho_i^{\text{init}} = \sigma(h_\theta(\mathbf{s}_i))$, where $\sigma(\cdot)$ is the sigmoid function, ensuring $\rho_i^{\text{init}} \in [0, 1]$. To capture interactions among slots, we further introduce an interaction matrix $\boldsymbol{W} \in \mathbb{R}^{K \times K}$, where each element $\boldsymbol{W}_{ij}$ is defined as $\boldsymbol{W}_{ij} = \text{softmax}_j\left((\boldsymbol{U}\mathbf{s}_i)^\top (\boldsymbol{V}\mathbf{s}_j)/\sqrt{d}\right)$. Here, $\boldsymbol{U}$ and $\boldsymbol{V}$ are learnable projection matrices, and $d$ is a scaling factor. The final importance score $\rho_i$ for each slot $\mathbf{s}_i$ is then computed as $\rho_i = \sum_{j=1}^{K} \boldsymbol{W}_{ij} \rho_j^{\text{init}}$. We then scale each slot representation by its respective importance score to create a weighted slot representation $\tilde{\mathbf{s}}_i = \rho_i \cdot \mathbf{s}_i$. To this end, the overall training objective of SSR can be formulated as:

$$\mathcal{L}_{\text{SSR}} = \|\hat{x} - x\|_2^2 + \lambda \sum_{i=1}^{K} \left\|\rho^{\text{init}}\right\|_2, \quad (3)$$

where $\lambda$ is the balancing parameter. The regularization term encourages sparsity in the initial importance scores, guiding the model to utilize only the most relevant slots for reconstruction. On the other hand, the less relevant slots are down-weighted rather than discarded, which may help retain subtle information. Compared to continuous semantic representations, this training process is more manageable due to the discretization of the slots. Since we lack truly fine-grained annotations, the learned slots or "concepts" may not be as readily interpretable as individual words. It is worth noting that incorporating a strong visual feature extractor could significantly boost baseline performance. However, for the sake of fairness, we have opted not to use approaches, such as DINOSAUR [64].

## 3.2 High-Order Relational Reasoning (HORR)

Albeit we have obtained part-based representations in the form of slots, their inner- and inter-object relations are still under-explored. Prior graph-based [9, 7] and non-graph [33] works only accommodate pairwise relationships (*e.g.*, connection between two nodes in a simple graph), but are inadequate to model the high-order relations inherent in images. Addressing this issue, we introduce a simple yet effective technique—HORR—to model high-order topological relations with a collection of graph nodes and hyperedges, divided into three stages: construction, learning, and matching.

**Definition 1 (Topological Homogeneity)** *We conceptualize topological homogeneity between the same object across images as a (hyper)graph matching problem, which is mathematically formulated as a relaxed quadratic assignment problem [52],*

$$\min_{\mathbf{X}} \|\mathbf{A} - \mathbf{X}\mathbf{B}\mathbf{X}^T\|_F^2 - tr(\mathbf{X}_u^T \mathbf{X}),$$
$$\mathbf{X} \in [0, 1]^{n \times m}, \mathbf{X}\mathbf{1}_n \leq \mathbf{1}_m, \mathbf{X}^T\mathbf{1}_m \leq \mathbf{1}_n, \quad (4)$$

*where $\mathbf{A} \in \mathbb{R}^{n \times n}$ and $\mathbf{B} \in \mathbb{R}^{m \times m}$ are the adjacent matrix encoding structure information of the graph $\mathcal{G}_A$ and $\mathcal{G}_B$ respectively, $n$ and $m$ are the number of graph nodes, $\|\cdot\|_F$ is the Frobenius norm, $\mathbf{X}_u \in \mathbb{R}^{m \times n}$ is the unary affinity matrix and generally specified as the node affinity $\mathbf{M}_{\text{aff}}$, and $\mathbf{X}$ is the relaxed permutation matrix encoding node-to-node assignment.*

**Hypergraph Construction.** Given visual slot representations from the previous step, we carry out a linear transformation to obtain graph nodes $\mathcal{V}_A$ and $\mathcal{V}_B$. This transformation is intended to map the visual features into the graph domain, ensuring effective graph matching. Then, we build hypergraph graphs $\mathcal{G}_A$ and $\mathcal{G}_B$ in both domains, modeling the topological structures inherent in the images, $x_A$ and $x_B$, respectively. The hypergraph [17] is defined as: $\mathcal{G}_{A/B} = (\mathcal{V}_{A/B}, \mathcal{E}_{A/B}, \mathbf{W})$, where $\mathcal{V}_{A/B}$ is the node sets, $\mathcal{E}_{A/B}$ is the hyperedge sets, and $\mathbf{W} \in \mathbb{R}^{|\mathcal{E}| \times |\mathcal{E}|}$ is a diagonal matrix of edge weights. The hypergraph is denoted by an incidence matrix $\mathbf{H} \in \mathbb{R}^{|\mathcal{V}| \times |\mathcal{E}|}$, where $\mathbf{H}(v,e) = 1$ indicates the node $v \in e$ and $\mathbf{H}(v,e) = 0$ indicates $v \notin e$. We adopt non-parametric density estimation, *i.e.,* K-Nearest Neighbors (KNN), to establish hyperedge connections. The Euclidean distance is used to calculate the distance between node embeddings.

**Hypergraph Learning.** After constructing the hypergraphs, we update the graph node features according to the connectivity defined by the hyperedges. Technically, we introduce hypergraph convolution [17] to perform message-passing and feature aggregation,

$$\tilde{\mathcal{V}} = \mathbf{D}_v^{-1/2} \mathbf{H} \mathbf{W} \mathbf{D}_e^{-1} \mathbf{H}^T \mathbf{D}_v^{-1/2} \boldsymbol{\Theta}, \tag{5}$$

where $\boldsymbol{\Theta}$ represents the parameter to be learned during training. By doing so, we can explicitly model intricate correlations among different parts of an image (*i.e.,* slots). On the other hand, in contrast to grids, sequences, and even graphs, this high-order modeling can more effectively capture relations among nodes, avoiding excessive or erroneous pairwise connections.

**Hypergraph Matching.** Since the graph node features have been updated by Eq. (5), we introduce an affinity matrix $\mathbf{W}_{\text{aff}}$ to measure the node correspondence between $\mathcal{G}_A$ and $\mathcal{G}_B$. Following [19, 44], we use the differentiable Sinkhorn layer [67] to calculate the affinity matrix $\hat{\mathbf{W}}_{\text{aff}}$, where each element indicates the degree of matching between pairs of nodes across graphs. To leverage the topological structures, we need a training objective to minimize the pairwise structural discrepancy between the hypergraphs [78]. In particular, we follow the Definition 1 to specify unary affinity matrix $\mathbf{X}_u$ as the obtained node affinity matrix $\hat{\mathbf{W}}_{\text{aff}}$. Since the constructed hypergraphs naturally encode rich high-order relationships, they enhance the cross-domain topological matching process. Conversely, the matching process introduces additional structural knowledge to the current graph through message propagation. Formally, the hypergraph-based matching objective is formulated as follows:

$$\mathcal{L}_{\text{mat}} = \underbrace{\sum_i \frac{1}{n} \left[ \max_j (\hat{\mathbf{W}}_{\text{aff}} \odot \mathbf{Y}_{\boldsymbol{\Pi}})_{i,j} - \mathbf{1} \right]^2}_{\text{enhance true positive matches}} + \underbrace{\sum_{i,j} \frac{1}{\|\mathbf{1} - \mathbf{Y}_{\boldsymbol{\Pi}}\|_1} \left[ \hat{\mathbf{W}}_{\text{aff}} \odot (\mathbf{1} - \mathbf{Y}_{\boldsymbol{\Pi}}) \right]^2_{i,j}}_{\text{penalize false positive matches}}, \tag{6}$$

where the $(i,j)$ element in $\mathbf{Y}_{\boldsymbol{\Pi}} \in \mathbb{R}^{n \times m}$ is $\mathbf{1}$ if $v_i^A \in \mathcal{G}_A$ and $v_j^B \in G_B$ belong to the same object class, otherwise it will be $\mathbf{0}$. Here, $\mathbf{Y}_{\boldsymbol{\Pi}}$ can be regarded as pseudo labels for matching due to the absence of ground-truth correspondence. Moreover, as the matching primarily targets high-level semantic relationships, we avoid imposing additional structural constraints [23, 43, 44] on this training objective, thereby significantly simplifying the training process.

### 3.3 Training and Inference

Putting everything together, the full training objective is formulated as follows:

$$\mathcal{L}_{\text{REMA}} = \mathcal{L}_{\text{CE}} + \alpha \mathcal{L}_{\text{SSR}} + \beta \mathcal{L}_{\text{mat}}, \tag{7}$$

where $\alpha$ and $\beta$ are hyper-parameters for balancing different loss terms. Our algorithm first trains deep models using the reconstruction objective to obtain discrete part representations with sufficient sparsity. Then it turns to model the cross-image components correlations as the graph matching problem where we further introduce the structural regularization term into the matching loss.

Note that our approach can be directly applied to test-time adaptation, similar to those self-supervised methods [69, 49, 3, 20]. The difference lies in the fact that their proxy tasks are somewhat heuristic, such as predicting rotations, jigsaw puzzles, and random masking. In contrast, our method directly exploits the intrinsic structured properties of the data, making it more robust and versatile.

Table 1: Comparison with OOD generalization methods on the PACS, Office-Home, and VLCS. *Note that the results reported in this table do not involve any test-time adaptation strategies.* Results are averaged over 3 random seeds. $\pm x$ denotes the rounded standard error.

| Algorithm | PACS | Office-Home | VLCS | Average Acc. (%) |
|---|---|---|---|---|
| **ERM** [72] | 85.5 | 67.6 | 77.5 | 76.7 |
| **CORAL** [68] | 86.2 | 68.7 | 78.8 | 77.9 |
| **DANN** [21] | 83.7 | 65.9 | 78.6 | 76.1 |
| **MLDG** [40] | 84.9 | 66.8 | 77.2 | 76.3 |
| **CDANN** [45] | 82.6 | 65.7 | 77.5 | 75.3 |
| **MMD** [42] | 84.7 | 66.4 | 77.5 | 76.2 |
| **IRM** [2] | 83.5 | 64.3 | 78.6 | 75.5 |
| **GroupDRO** [62] | 84.4 | 66.0 | 76.7 | 75.7 |
| **I-Mixup** [80, 82, 84] | 84.6 | 68.1 | 77.4 | 76.7 |
| **RSC** [28] | 85.2 | 65.5 | 77.1 | 75.9 |
| **ARM** [88] | 85.1 | 64.8 | 77.6 | 75.8 |
| **MTL** [5] | 84.6 | 66.4 | 77.2 | 76.1 |
| **VREx** [36] | 84.9 | 66.4 | 78.3 | 76.5 |
| **Mixstyle** [95] | 85.2 | 60.4 | 77.9 | 74.5 |
| **SelfReg** [32] | 85.6 | 67.9 | 77.8 | 77.1 |
| **SagNet** [55] | 86.3 | 68.1 | 77.8 | 77.4 |
| **GVRT** [53] | 85.1 | 70.1 | 79.0 | 78.1 |
| **VNE** [33] | 86.9 | 65.9 | 78.1 | 77.0 |
| **REMA (Ours)** | $88.7_{\pm0.3}$ | $72.0_{\pm0.4}$ | $79.4_{\pm0.3}$ | **80.0** |

## 4 Experiments

In this section, we empirically evaluate the proposed REMA in two types of OOD scenarios, *i.e.,* OOD generalization and test-time adaptation. In the following, we first describe the experimental setup (Section 4.1) and then provide the main results (Section 4.2) and ablation studies (Section 4.3).

### 4.1 Experimental Setup

**Datasets.** For OOD generalization, we leverage the three most widely used benchmark datasets. **PACS** [39] comprises 9,991 images and exhibits significant variations in image styles. It consists of 4 domains each with 7 classes, *i.e.,* Photo, Art Painting, Cartoon, Sketch. **VLCS** contains 10,729 images of 5 classes from 4 photographic domains: PASCAL VOC 2007, LabelMe, Caltech, Sun. **Office-Home** [73] is collected from both office and home environments, and its domain shifts stem from variations in viewpoint and image style. It has 15,500 images of 65 classes from 4 domains, *i.e.,* Artistic, Clipart, Product, Real World. Regarding test-time adaptation, we follow the common benchmarks [75, 30, 79] that utilize **CIFAR-10/100** [35] and **ImageNet** [14] as the ID (training) data. **CIFAR-10/100C** [26] and **Imagenet-C** [26] are used as OOD (test) data, comprising different corruptions applied to their original datasets.

**Implementation Details.** For OOD generalization, we use ResNet-50 for PACS, Office-Home, and VLCS and ResNet-18 for CIFAR-10. The model is trained using stochastic gradient descent with momentum 0.9, and weight decay $10^{-4}$. The training batch size is set to 128. The learning rate is $10^{-4}$. Following common practice, the model selection is based on a training domain validation set. For test-time adaptation, we use ResNet-50 for all datasets. We utilize the Adam optimizer to update the network parameters. To facilitate a fair comparison, the test batch size of 64 in all methods. $\lambda$ in Eq. (3) is set to 0.01 in all experiments. $\alpha$ and $\beta$ in Eq. (7) is set to 10 and 0.1, respectively.

**Baselines.** We compare REMA against two types of baseline methods. **(1) OOD Generalization:** We adopt the leave-one-domain-out evaluation protocol and follow the model selection strategy used in [25]. The main baselines have optimization-based CORAL [68], MLDG [40], CDANN [45] and MMD [42], augmentation-based Mixstyle [95], SagNet [55] and I-Mixup [80, 82, 84], and so on. **(2) Test-time adaptation:** All methods are based on online batch-level test data adaptation setting and the following baselines are included: entropy-minimization: Tent [75] and SHOT [47], pseudo-labeling: T3A [29] and TAST [31], consistency-alignment: TSD [79] and TIPI [56].

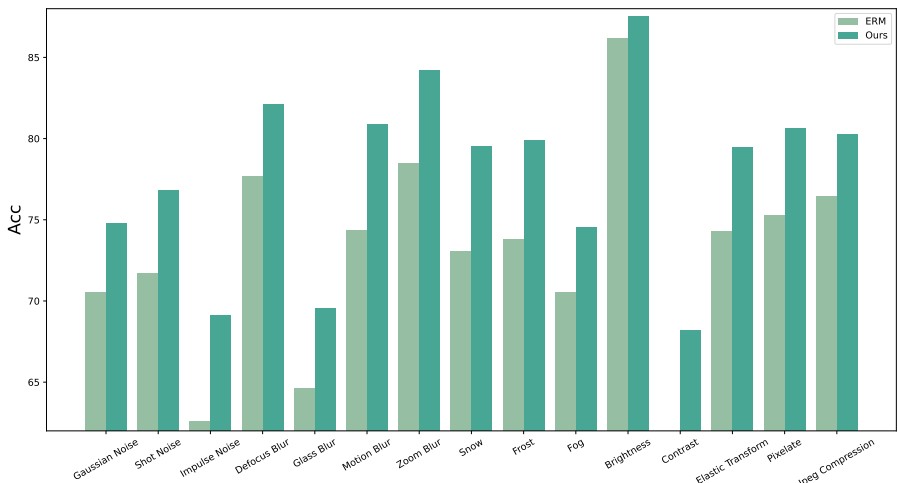

Figure 3: Generalizing from CIFAR-10 to CIFAR-10C using ERM and our REMA.

Table 2: Comparisons with the state-of-the-art methods with average error rate (%) on image corruption benchmarks. Testing is conducted on the highest level of image corruption. All methods use ResNet-50 backbone. ↓ means lower is better.

| Method | CIFAR-10C ↓ | CIFAR-100C ↓ | ImageNet-C ↓ | Avg. ↓ |
|---|---|---|---|---|
| No Adaptation | 29.1 | 60.4 | 82.0 | 57.2 |
| +SHOT [47] | 15.3 | 41.5 | 58.3 | 38.4 |
| +Tent [75] | 14.0 | 39.0 | 58.1 | 37.0 |
| +PL [37] | 22.3 | 40.1 | 63.0 | 41.8 |
| +T3A [29] | 26.7 | 58.3 | 75.8 | 53.6 |
| +TAST [31] | 26.6 | 60.7 | - | - |
| +TAST-BN [31] | 13.1 | 37.8 | 67.1 | 39.3 |
| +TIPI [56] | 13.5 | 38.3 | 55.9 | 35.9 |
| +TSD [79] | 13.1 | 37.7 | 53.2 | 34.6 |
| +REMA | 12.0 | 35.8 | 52.2 | **33.3** |

## 4.2 Main Results

**OOD Generalization.** The main results are presented in Table 1 and Figure 3. Notably, REMA consistently outperforms all baseline methods by a large margin in each dataset. For example, compared to the recent method VNE, REMA improves classification accuracy by 1.8% for PACS, 6.1% for Office-Home, and 1.3% for VLCS, revealing the importance of our reconstruction (semantic abstraction) and matching (topological homogeneity) modules. Moreover, there are two notable observations: (1) Compared to statistical matching methods (*e.g.,* CORAL, DANN, and MMD) that directly optimize moment matching objectives in the latent space, REMA demonstrates superior performance by introducing topological information with more sparse input (slots). (2) Style or data augmentation based methods (*e.g.,* MixStyle) is simple and easy-to-implement. However, their intuitive nature of interpolating around the training set may not accurately cover the target distribution region. By contrast, REMA directly learns major components from training data. (3) Compared to PACS and VLCS datasets, the Office-Home dataset has more categories and total samples. Many baseline approaches even perform worse than ERM, indicating their limited scalability. Instead, REMA demonstrates a larger performance gain in this challenging task.

**Test-Time Adaptation.** We compared REMA with existing advanced TTA methods, and the results are summarized in Table 2 and Table 3. REMA consistently provides improvements on multiple datasets. Compared to TSD [79], it achieves a 1.3% improvement on three pixel-level corruption datasets and a 3.1% improvement on three challenging cross-domain datasets. Compared to the SOTA, REMA has three major advantages: (1) Tent [75] or SHOT [47] based on entropy minimization has limitations, as they can overcome distribution shifts caused by pixel corruption but fail to handle cross-domain test data, even leading to performance degradation. However, REMA can handle both types of OOD data. (2) T3A [29] and TAST [31] based on pseudo-labels have performance limitations, and the pros and cons of pseudo-labels significantly impact the adaptation, which can be

Table 3: Comparisons with the state-of-the-art methods on three image classification benchmarks.

| Method | VLCS | PACS | Office-Home | Average Acc. (%) |
|---|---|---|---|---|
| ERM | $76.7_{\pm0.5}$ | $83.2_{\pm1.1}$ | $67.1_{\pm1.0}$ | 75.3 |
| +Tent [75] | $73.0_{\pm1.3}$ | $85.2_{\pm0.6}$ | $66.3_{\pm0.8}$ | 74.9 |
| +TentClf [75] | $75.8_{\pm0.7}$ | $82.7_{\pm1.6}$ | $66.8_{\pm1.0}$ | 75.1 |
| +SHOT [47] | $67.1_{\pm0.9}$ | $84.1_{\pm1.2}$ | $67.6_{\pm0.7}$ | 72.9 |
| +T3A [29] | $77.3_{\pm0.4}$ | $83.9_{\pm1.1}$ | $68.3_{\pm0.8}$ | 76.5 |
| +TAST [31] | $77.7_{\pm0.5}$ | $84.1_{\pm1.2}$ | $68.6_{\pm0.7}$ | 76.8 |
| +TAST-BN [31] | $73.5_{\pm1.4}$ | $89.2_{\pm0.5}$ | $68.9_{\pm0.5}$ | 77.2 |
| +TSD [79] | $74.5_{\pm0.9}$ | $89.3_{\pm0.6}$ | $68.4_{\pm0.7}$ | 77.3 |
| +REMA | $79.4_{\pm0.4}$ | $90.3_{\pm0.3}$ | $71.6_{\pm0.6}$ | **80.4** |

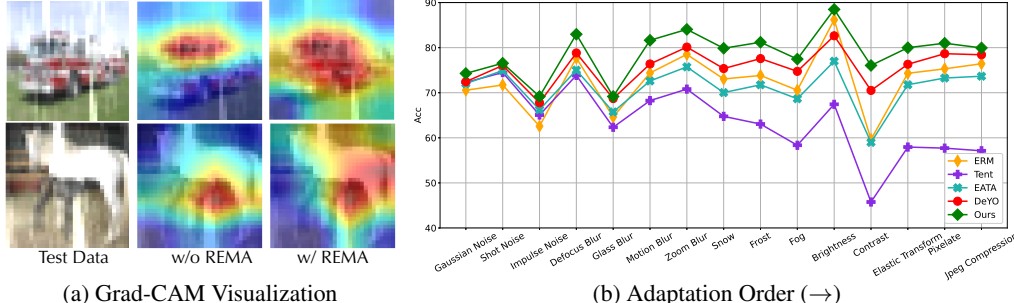

| Test Data | w/o REMA | w/ REMA |
|---|---|---|

(a) Grad-CAM Visualization      (b) Adaptation Order ($\rightarrow$)

Figure 4: **(a)** Visualization. **(b)** Analysis on continuous test-time adaptation.

observed in their ceilings on two types of datasets. REMA enhances resistance to noisy labels by capturing higher-order semantic dependencies. (3) REMA converts dense pixels into sparse slots, accurately capturing domain-invariant features, which will not be affected by pixel corruption and domain shift. Implementing consistent optimization during test time makes REMA more robust, resulting in a 5.1% accuracy improvement across all benchmarks, *i.e.,* ERM *vs.* REMA.

### 4.3 Ablation Studies

**Ablations of key modules in REMA.** In this part, we provide the ablation results in Table 4, investigating the independent and combined effects of SSR and HORR proposed in REMA. As can be seen, incorporating SSR and HORR separately leads to improved generalization performance, demonstrating their contributions to abstraction and relational modeling. In addition,

Table 4: Ablation of REMA (%).

| ASR | HORR | VLCS | PACS | Office-Home |
|---|---|---|---|---|
| × | × | 76.7 | 83.2 | 67.1 |
| ✓ | × | 78.6 | 88.1 | 70.3 |
| × | ✓ | 78.3 | 89.0 | 70.2 |
| ✓ | ✓ | 79.4 | 90.3 | 71.6 |

integrating SSR and HORR in our method yields the best performance, highlighting that the two modules systematically work together and reciprocate each other. Moreover, as shown in Figure 4(a), the full REMA can provide more complete and accurate representations of objects.

**Analysis on continuous test-time adaptation.** We empirically evaluate the continuous learning ability of REMA by comparing it with state-of-the-art test-time adaptation methods, namely Tent [75], EATA [57], and DeYO [38]. The results are presented in Fig. 4(b), where the adaptation order is from left to right. We can see that the proposed REMA substantially and consistently outperforms all baseline methods as adaptation proceeds, revealing the robustness and the capability to handle open-world changes.

**Analysis on hypergraph matching.** In Figure 5, we visualize the learned doubly stochastic node affinity matrix and the ground-truth (GT) matrix. We observe that the proposed graph matching method effectively identifies the correct node affinity, revealing its effectiveness in handling cross-domain higher-order relationships.

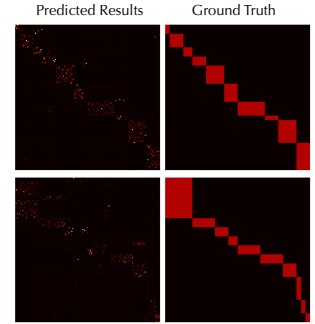

Predicted Results    Ground Truth

Figure 5: Learned affinity *vs.* GT

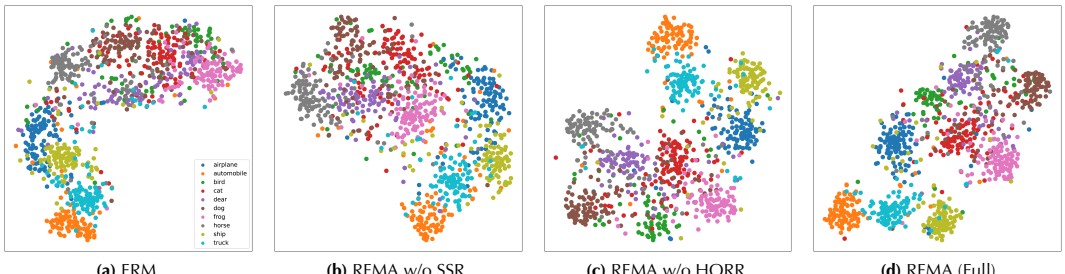

Figure 6: Feature visualization of different methods using $t$-SNE.

**Feature visualization.** Figure 6 demonstrates the $t$-SNE [71] visualization of feature embeddings for ERM, REMA w/o SSR, REMA w/o HORR, and REMA (Full). We extract feature embeddings using CIFAR-10C data, where the corruption type is snow with a severity level of 5. The different color stands for different classes. We can observe that our REMA exhibits better clustering patterns with both SSR and HORR modules. When SSR is removed, some closely related categories (*e.g.,* animals) become highly mixed, indicating the advantage of SSR in extracting robust semantic representations. On the other hand, without HORR, intra-class variations increase, indicating some degree of ambiguity between different categories. Our HORR addresses semantic ambiguity by modeling topological homogeneity.

## 5 Related Work

**OOD Generalization.** The aim of OOD Generalization is to train a model using data from the source distribution so that it can perform well on an unknown target distribution. A series of works can be divided into three categories: (1) Minimizing distribution differences [42, 45, 68, 90]: using meta-learning to simulate distribution shift [40, 18], or learning an invariant transformation based on adversarial learning [42, 45, 89, 94, 93]. (2) Domain-invariant representation learning: by analyzing the feature learning process of deep neural networks [16, 66, 27, 63, 12, 13], it is inferred how ID data can be generalized to OOD data [6, 65]. By understanding the interactions between ERM and generalization, domain-invariant features are learned. (3) More generic OOD generalization: many studies focus on generalization under a relaxed assumption, such as open environments with unknown classes [8], semi-supervised settings in the testing environment [86], adapting foundation models [91], and OOD data synthesis [60, 59, 22], aiming to identify generalizable features from sophisticated test distributions. However, previous studies have only focused on the instances themselves and have not explored the inherent high-order semantic dependencies in ID and OOD data, overlooking the topological homogeneity between distributions.

**Test-time Adaptation.** The goal of TTA [46] is to adjust the source trained model using test data without ground truth during the testing phase [47, 58, 10, 87, 79, 29]. Most of the existing methods directly perform gradient optimization on the test samples, with optimization objectives including prediction entropy [75], self-training and stochastic restoring [77], or based on normalization [54], etc., making the model adapt to the dynamic target environment at each step. Recently works further consider the scenario of noisy outlier samples [24, 70] appearing in the test stream. However, previous studies have simply optimized the test instances based on self-training, ignoring the topological relationship between samples and the high-order semantic dependencies within the test data steam, leading to suboptimal adaptation.

## 6 Conclusion

In this paper, we propose to achieve out-of-distribution robustness from the perspective of ensuring topological homogeneity between the same object class regardless of the surrounding environments. To achieve this goal, we propose a new framework REMA to first obtain a set of sparse and discrete representations from dense pixels and then model the high-order topological relations and dependencies via hypergraph (a generalized form of graph). Our experimental results reveal that REMA achieves superior performance on standard OOD generalization and test-time adaptation benchmarks. In the future, we aim to extend this framework to more complex and dynamically changing scenes.

## Acknowledgement

We thank the reviewers for their valuable comments. This work was supported in parts by NSFC (U21B2023, U2001206), ICFCRT(W2441020), Guangdong Basic and Applied Basic Research Foundation (2023B1515120026), DEGP Innovation Team (2022KCXTD025), Shenzhen Science and Technology Program (KQTD20210811090044003, RCJC20200714114435012), and Scientific Development Funds from Shenzhen University.

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
