# OpenReview forum: "Reconstruct and Match: Out-of-Distribution Robustness via Topological Homogeneity"
_NeurIPS.cc/2024/Conference — NeurIPS 2024 spotlight_

### Official Review · Reviewer_EaaQ · 2024-07-11

**Soundness:** 3
**Presentation:** 3
**Contribution:** 3
**Rating:** 6
**Confidence:** 2

**Summary:**

The paper proposes a method for improving domain generalization and test time adaptation by introducing a selective variant of slot attention, formulating the relationship between slots across images as topological homogeneity between hypergraphs constructed based on the slots, and thereby matching occurances of the same object between different images.

**Strengths:**

The proposed method and its components introduce novel techniques for the domain generalization problem that are well motivated and appear to be sound.

The mathematical derivation of the employed forms of the individual steps makes sense.

Several ablations well demonstrate the dependence of REMA's results on its main building blocks.

**Weaknesses:**

The proposed method consists of fairly many newly introduced steps.
While ablations show their necessity of the major modular building blocks of REMA for achieving the eventual results, these building blocks are not analyzed in detail.
For understanding them (and therefore REMA), it would be necessary to expound each individual step and confirm that it has the intended effect and that the explicit or implicit mathematical assumptions hold.
For example, the results of standard slot attention can be of varying quality, and it would be important to verify whether and how such imprecisions introduced there propagate through the next steps.
The analysis in Figure 5 goes in this direction, but it is not clear what exactly is probed and measured here. A similar analysis as in for the other components of the method would be helpful.

**Main issue of the paper**: There exist different communities in ML which study the generalization performance of DNNs to distribution shifts in very different settings. Unsupervised Domain Adaptation (UDA) is separate from Domain Generalization (DG) and both are separate from OOD generalization. Putting papers which study either of those in the same Table is wrong, misleading and highly confusing. All of these settings have their benefits and challenges and the numbers are simply incomparable if the initial conditions, such as which data is available during training, differ this strongly. The authors conflate and confuse all three settings on multiple occasions which makes the paper confusing and the results are impossible to understand. It is further completely unclear in which of the three their proposed method falls into. Below, I explain the issue in much more detail and provide specific problematic text instances. The paper needs a major revision which includes reworking all sections and most Tables.

line 183: "For OOD generalization, we leverage the three most widely used benchmark datasets." The datasets PACS and Office-Home are usually used in Unsupervised Domain Adaptation (UDA), not OOD generalization. Datasets commonly used to benchmark OOD generalization are datasets like ImageNet-R, ImageNet-Sketch, ObjectNet or ImageNet-V2. The choice of the baselines the authors compare to also gives the impression that they confuse domain adaptation with OOD generalization: CORAL and DANN are UDA methods.


line 66: "The goal of OOD generalization is to find a predictor f : X 7→ Y that generalizes well to all unseen target domains." This is correct, but then the used baselines are wrong because they assume access to the target domains and are trained using this information. It is not clear to me how the UDA methods (shown in Table 1) can be implemented without this access.


The setting of UDA differs drastically from OOD generalization: In UDA, we attempt to learn a model which performs well on the source domain and a set of target domains, using unlabeled data from the target domains. That is, in UDA, we assume access to the unlabeled test set at training time. This differs from the OOD generalization setting: Here, we want to train a model on a source dataset and then test its generalization to an unseen test set. That is, we assume no knowledge about the test time distribution shift. This is a crucial difference and it is wrong to confuse the two terms. The implemented baselines, e.g. CORAL and DANN, are used in a UDA setup, not in an OOD generalization setup. In this light, the wording "Following common practice, the model selection is based on a training domain validation set" is meaningless because the common practice for model selection differs depending on which setting we consider.


The related work section is similarly confusing. On the one hand, the authors wish to review works on OOD generalization where the goal is "to train a model using data from the source distribution so that it can perform well on an unknown target distribution". [41] and [42] use the DG setting, i.e. assumes access to n-1 target domains. [63] assumes access to all target domains. It is unclear why [61] is a good citation for domain-invariant learning. I haven't checked all citations, but the ones I checked **all** assumed access to the target domains.


The setting the baselines use in Table 1 is inconsistent. CORAL and DANN have been trained in an UDA setting. That is, they assume access to unlabeled target data at test time from a target domain. VNE and IRM are used in a domain generalization setting where they assume access to n-1 domains and aim to perform well on an unseen domain. These are two very different settings which are incomparable to each other and it is wrong and confusing to put them in the same table without any discussion. Further, the authors aim to test OOD generalization where no access to unseen target domain data is available (line 66) which is incompatible with either UDA or domain generalization.


The baseline numbers for the cited papers are wrong. I checked the numbers of CORAL and VNE from Table 1 with the numbers in the original papers and they do not match. If the authors reimplemented all the baselines, they need to state this and discuss where the performance differences come from.


Checking papers-with-code for the PACS dataset (https://paperswithcode.com/sota/domain-generalization-on-pacs-2), the best numbers are close to 100% for the domain generalization setting. Since the authors use DG benchmarks in their table 1, I assume that using other benchmarks in the DG domain would be valid as well. If we filter for RN50 architectures only, the best number is 90.5% which is higher compared to the best number reported by the authors.


Looking at the Office-Home dataset (https://paperswithcode.com/sota/unsupervised-domain-adaptation-on-office-home), the best number here is 90%. Here, the setting is UDA. But as mentioned before, it is not clear which setting should be used because the authors include both UDA and DG benchmarks in their Table 1. The best number in the UDA setting is higher than what the authors report in Table 1. There are actually five separate benchmarks for OfficeHome on papers with code (Domain Generalization, Domain Adaptation, Universal Domain Adaptation, UDA, Partial DA) and it is not clear which one should be used.


It is not clear how the models are trained. The test sets are comprised of different domains. Do the authors train their models on all of them or on e.g. two of the domains and test against the rest or on one of the domains and then test against the rest?


For test-time adaptation, the authors missed important works which perform better than their method. SLR achieves an error of 48.7% on the highest severity of ImageNet-C [A]. ETA and EATA [B] achieve an error of around 52%.
[A] Mummadi et al. "Test-time adaptation to distribution shift by confidence maximization and input transformation"
[B] Niu et al. "Efficient test-time model adaptation without forgetting."


Press et al. [C] showed that most test-time adaptation methods collapse when adapting for long periods of time. The authors should test their model on the proposed CCC benchmark to analyze whether their method also suffers from the collapse. The authors are writing that they test their method in  the continuous test-time adaptation scenario in Fig. 4b, but it is entirely unclear on which dataset they are testing it. Is it the CCC benchmark? Notably, if the authors are testing their approach on the Continual Test-time adaptation benchmark from [D] (although I am just guessing here), Press et al. showed in [C] that one needs to adapt for longer time periods to show-case collapse and the adaptation periods proposed in [D] are insufficient. The authors must state clearly which continual learning benchmark they are using and I would suggest to report numbers on both [C] and [D].
[C] Press et al. "RDumb: A simple approach that questions our progress in continual test-time adaptation"
[D] Wang et al. "Continual test-time domain adaptation"

**Questions:**

The paper requires a major revision to make it much more clear which setting the authors are targetting: Is it:
- OOD generalization as suggested by the title? Then, the authors **cannot** assume any knowledge about the distribution shift at test time. Then the baselines in Table 1 cannot be used as they do assume access to the target domains.
- Unsupervised Domain Adaptation as suggested by the baselines in Table 1, i.e. DANN and CORAL? Then, the authors need to reformulate their title, abstract, motivation etc. and remove all instances of "OOD generalization" because this is very confusing otherwise. The authors also need to remove the Domain Generalization baselines from Table 1.
- Domain Generalization as suggested by the IRM and VNE baselines in Table 1? Then, the authors need to remove the UDA baselines from their Table 1 and remove instances of mentioning OOD generalization from the paper.


The authors need to add Test-Time adaptation baselines to their Table and improve their continual Test-Time evaluation.

The authors need to make more comprehensive evaluations on the influence of the different components of their loss.

**Limitations:**

The limitations section in the appendix is very short. A clear limitation is that the method introduces several hyperparameters which need tuning. This imposes an additional computational overhead which has not been discussed.

Another limitation which can be fixed is that the authors did not study the continual learning setting to analyze whether their method collapses when adapting for long time periods.

In 2024, most people work with large-scale pretrained models such as e.g. CLIP. It is not clear how this method could be used on those models and whether improving ResNet50's performance is relevant nowadays.

---

> ### Author Rebuttal · Authors · 2024-08-06
>
> We greatly appreciate the time and effort the reviewer has invested in reviewing our paper. However, **we must point out that the review contains many factual errors and misunderstandings, rendering most comments unacceptable due to their erroneous assumptions.** We will try our best to eliminate the misunderstandings via the following responses.
>
> > **Q1: Regarding problem definitions and experimental comparisons**
>
> (1) Our paper is NOT related to the UDA setting; we consistently emphasize our research on OOD generalization (a.k.a. domain generalization). The reviewer's comment that "Unsupervised Domain Adaptation (UDA) is separate from Domain Generalization (DG) and both are separate from OOD generalization" is **problematic**. In our paper, **DG and OOD generalization are synonymous and can be used interchangeably**. This is also a **consensus** within the community, as exemplified by two well-known DG surveys [1,2] that treat the terms as synonymous. In the abstract of [1], it is stated: "Domain generalization (DG), i.e., out-of-distribution generalization, has attracted increasing interest in recent years."
>
> (2) In Table 1, we ONLY compare with DG baselines. The reviewer's claim that our baseline comparisons, which include DANN and CORAL, suggest we are evaluating UDA algorithms is a **significant misunderstanding**. DANN and CORAL, originally introduced as UDA methods, are now important DG baselines that match deep features from different source domains (NOT target domain) via domain adversarial training (DANN) and correlation alignment (CORAL). This is a **common practice in the DG community**. The well-known **DomainBed** benchmark (code: https://github.com/facebookresearch/DomainBed) and many follow-up works in the field also include both DANN and CORAL as their main DG baselines. Regarding the baseline results, we cite all these numbers from DomainBed or their original papers (following the identical setting) to ensure a fair comparison. Our experiments were also conducted in the same conditions. Please refer to recent papers to avoid misunderstandings regarding benchmarks, such as [3-6]. Based on this, we need to emphasize that the comparisons in Table 1 are reasonable and fair, and our problem definition is also precise and clear.
>
> (3) The reviewer's statement that ''The baseline numbers for the cited papers are wrong. I checked the numbers of CORAL and VNE from Table 1 with the numbers in the original papers and they do not match'' is **incorrect**. Regarding CORAL, its original paper conducted UDA (not DG) experiments on Office-31. **Moreover, it did not involve any experiments on PACS, Office-Home, or VLCS at all. Why then is it claimed that our ''cited'' results are wrong?** Regarding VNE, we used the results from Table 2 of the original paper, which were obtained using the **ERM** algorithm (see our Table 1). The results from Table 3 of the original VNE paper, which were obtained using the **SWAD** algorithm, are inconsistent with our experiments and thus cannot be used. Regarding the mentioned results from paperswithcode website, it is not standard practice in the DG community to base comparisons on them due to the potential distinction in training and inference. For example, the paper that achieved 90.5% on PACS employed a different model selection strategy (not ID val).
>
> (4) The reviewer's statement that ''the PACS and Office-Home datasets are usually used in Unsupervised Domain Adaptation (UDA) and not OOD generalization'' is **incorrect**. **The PACS dataset was introduced in the paper "Deeper, Broader and Artier Domain Generalization," which is a DG paper, not UDA.** Actually, the datasets we use are among the most commonly utilized in the DG community for image data, as referenced in [1-6].
>
> (5) We clarify why our paper also conducted experiments related to test-time adaptation (TTA). The goal of TTA [7] is to update the model online during testing, which complements the goal of DG, i.e., DG aims to learn a generalizable model using only source data, while TTA seeks to enhance generalization ability using unlabeled target data. On the other hand, a series of TTA works utilizes self-supervised learning tasks for both the training and testing phases (L174-177). This aligns with our slot-based approach, and our hypergraph-based matching naturally links training and test phases, enabling our framework to also work as a TTA method. We reiterate that the experiments for DG and TTA were conducted separately (Tab. 1 does not utilize any TTA strategies).
>
> **References**
>
> [1] Generalizing to Unseen Domains: A Survey on Domain Generalization. TKDE, 2022.
>
> [2] Domain Generalization: A Survey. TPAMI, 2022.
>
> [3] Diverse Weight Averaging for Out-of-Distribution Generalization. NeurIPS, 2022.
>
> [4] Improving Out-of-Distribution Generalization by Adversarial Training with Structured Priors. NeurIPS, 2022.
>
> [5] MADG: Margin-based Adversarial Learning for Domain Generalization. NeurIPS, 2023.
>
> [6] On the Adversarial Robustness of Out-of-distribution Generalization Models. NeurIPS, 2023.
>
> [7] A Comprehensive Survey on Test-Time Adaptation under Distribution Shifts. IJCV, 2024.
>
> > **Q2: Regarding TTA experiments**
>
> (1) We thank the reviewer for pointing out these two works [A, B] and have added them to our paper. However, we have to note that SLR [A] is still a preprint paper (since 2021), uses extra input augmentation (compared to other baselines), and does not release code (hard to make a fair comparison).
>
> (2) Regarding continuous TTA, we have already presented it in Fig. 4(b) (CIFAR-10C), where the x-axis represents the type of corruption and multiple SOTA TTA methods are compared. More results (CIFAR-100C and ImagNet-C) will be added to the final version.
>
> > **Q3: Regarding ablation studies**
>
> In the original manuscript, the ablation of REMA is shown in Tab. 4, Fig. 4, and Fig. 6. As suggested, we provide additional ablation studies in **the attached PDF (Tab. 2)**.

---

> ### Comment · Reviewer_EaaQ · 2024-08-10
> **I apologize for the misunderstandings in my review**
>
> Dear authors,
>
> I would like to acknowledge major misunderstandings of the paper and the relevant literature on my part. I was not aware that DANN and CORAL are now also DG benchmarks. Seeing those in the table contributed to me thinking the paper is based in UDA. I apologize for missing this and will raise my score to 6 and withdraw my concerns.
>
> Best, reviewer Eaaq

---

> > ### Author Response · Authors · 2024-08-10
> > **Appreciation for Your Revised Review and Understanding**
> >
> > Dear Reviewer EaaQ,
> >
> > Thank you for your thoughtful reconsideration of our paper. We appreciate your willingness to update your evaluation and withdraw your previous concerns. Your revised understanding and increased score significantly aid in the review process, and we are grateful for your efforts to resolve these issues.
> >
> > Best regards,
> >
> > Submission4013 Authors

---

### Official Review · Reviewer_rhAB · 2024-07-12

**Soundness:** 2
**Presentation:** 2
**Contribution:** 2
**Rating:** 5
**Confidence:** 2

**Summary:**

This paper presents REMA which designed to improve the robustness of deep learning models against out-of-distribution (OOD) data. REMA employs a selective slot-based reconstruction module to dynamically map dense pixels into a sparse set of slot vectors, enabling the identification of major components from objects in an unsupervised manner. Additionally, a hypergraph-based relational reasoning module is introduced to model high-order dependencies among these components, ensuring topological homogeneity. Experiments conducted on standard benchmarks demonstrate that REMA outperforms state-of-the-art methods in OOD generalization and test-time adaptation settings, highlighting its effectiveness in handling distribution shifts and enhancing the adaptability of deep models in real-world, non-stationary environments.

**Strengths:**

1. REMA introduces a unique combination of selective slot-based reconstruction and hypergraph-based relational reasoning to address OOD robustness, which has not been explored extensively in previous studies.

2. The framework effectively identifies and leverages major components of objects without requiring human prior knowledge or fine-grained annotations, reducing the need for extensive labeled data.

3. The paper provides extensive experimental results on multiple benchmark datasets, showing improvements over existing state-of-the-art methods in both OOD generalization and test-time adaptation scenarios.

**Weaknesses:**

1. The paper does not provide a detailed analysis of the computational overhead introduced by the new modules, which could impact the scalability of the approach for large-scale or real-time applications.

2. While the experiments cover several benchmark datasets, the generalizability of REMA to other types of datasets or more diverse real-world scenarios is not fully explored.

3. The effectiveness of REMA relies on several hyperparameters (e.g., the number of slots, attention iterations), and the paper does not thoroughly investigate the sensitivity of the model's performance to these parameters.

**Questions:**

1. Parameter Sensitivity: How sensitive is REMA's performance to the choice of hyperparameters such as the number of slots and attention iterations? Could an automated hyperparameter tuning method improve the robustness and generalization of the model further?

2. Computational Efficiency: What is the computational overhead associated with the selective slot-based reconstruction and hypergraph-based relational reasoning modules? How does this overhead impact the scalability and real-time applicability of REMA in large-scale or resource-constrained environments?

**Limitations:**

Yes

---

> ### Author Rebuttal · Authors · 2024-08-06
>
> We thank the reviewer for the constructive feedback, which we address below:
>
> > **Q1-1: How sensitive is REMA's performance to the choice of hyperparameters such as the number of slots and attention iterations?**
>
> As suggested, we have provided quantitative experimental results in **the attached PDF (Fig. 3)**. Within a reasonable range, REMA is *not sensitive* to changes in hyperparameters (*max variation: ~1.8%*). Note that increasing the number of slots and iterative attention times will lead to higher computational costs.
>
> > **Q1-2: Could an automated hyperparameter tuning method improve the robustness and generalization of the model further?**
>
> In our main paper, to ensure fairness in comparisons, we strictly follow the traditional DG methods for model selection. Introducing automated hyperparameter tuning could alter the experimental benchmarks. Moreover, searching a set of hyperparameters for each task would increase the training cost. As demonstrated in Q1-1, we empirically found that the proposed method is not sensitive to changes in hyperparameters. Therefore, we opted to fix hyperparameters across all experiments. On the other hand, it is true that this may not be the optimal solution for individual cases. Leveraging hyperparameter searching techniques to identify potentially optimal parameters for each case could further enhance the final performance.
>
> > **Q2-1: What is the computational overhead associated with the selective slot-based reconstruction and hypergraph-based relational reasoning modules?**
>
> In **the attached PDF (Fig. 4 and Fig. 5)**, we have demonstrated the computational overhead of the selective slot-based reconstruction (SSR) and hypergraph-based relational reasoning (HORR) modules. (1) During training, compared to previous OOD generalization methods, our SSR, which utilizes a small number of slots, does not significantly increase computational costs; HORR also does not incur substantial overhead due to the limited number of hypergraph nodes and edges. Thus, our method (SSR + HORR) is close to previous methods in terms of param. and GFLOPs **(Fig. 4)**. (2) During inference, the speed of our method is comparable to most previous TTA methods **(Fig. 5)**. Note that without TTA, it is equivalent to the inference speed of ERM.
>
> > **Q2-2: How does this overhead impact the scalability and real-time applicability of REMA in large-scale or resource-constrained environments?**
>
> (1) Scalability. First, the time complexity of Slot Attention is approximately O(N×K×d), which indicates that the computational cost of the algorithm increases linearly with the increase in the length of the input sequence, the number of slots, and the feature dimensionality. In our case, N and K are generally O(1), and d is typically O(10^2), thus the complexity of hypergraph convolution is approximately O(10^2). Second, for hypergraph convolution, the time complexity is often in the order of O(E×C×d), where E is the number of hyperedges, C is the average cardinality of the hyperedges (i.e., the average number of vertices that each hyperedge connects), and d is the dimensionality of the features. Similarly, E and C are generally O(1), and d is typically O(10^2), thus the complexity of hypergraph convolution is approximately O(10^2). Based on this detailed analysis of complexities, we find that the two main operations in REMA are efficient, indicating the potential to scale up.
>
> (2) Real-time applicability. In our main paper (Sec. 4.3 and Fig. 4(b)), we show the results of continuous test-time adaptation, which aims to evaluate the capability to handle dynamically changing environments. Our method consistently outperformed SOTA methods, showing superior stability without significant changes. As mentioned in Q2-1, our computational overhead is not substantial. Therefore, combined with our capability for continuous adaptation, this ultimately allows us to meet real-time demands—ensuring both speed and stability.
>
> > **Q3: The generalizability of REMA to other types of datasets or more diverse real-world scenarios.**
>
> Thank you for your suggestion. As suggested, we have added two typical applications in medical scenarios, including pneumonia classification (chest X-ray images) [1] and skin lesion classification [2]. Please refer to the original papers for details of the experimental setup. The results are presented below.
>
> **Pneumonia Classification.** Chest X-ray images from three different sources: NIH, ChexPert, and RSNA. The task is to detect whether the image corresponds to a patient with Pneumonia or not.
>
> | Method      | RSNA | ChexPert | NIH  |
> | ----------- | ---- | -------- | ---- |
> | ERM         | 55.1 | 60.9     | 53.4 |
> | IRM         | 57.0 | 63.3     | 54.6 |
> | CSD         | 58.6 | 64.4     | 54.7 |
> | MatchDG [1]   | 58.2 | 59.0     | 53.2 |
> | MiRe [3]    | 63.6 | 65.0     | 56.4 |
> | REMA (Ours) | 68.2 | 70.5     | 62.4 |
>
> **Skin Lesion Classification.** We adopt seven public skin lesion datasets, including HAM10000, Dermofit (DMF), Derm7pt (D7P), MSK, PH2, SONIC (SON), and UDA, which contain skin lesion images collected from different equipment.
>
> | Method      | DMF  | D7P  | MSK  | PH2  | SON  | UDA  | Avg  |
> | ----------- | ---- | ---- | ---- | ---- | ---- | ---- | ---- |
> | DeepAll     | 24.9 | 56.8 | 66.7 | 80.0 | 86.1 | 62.6 | 62.9 |
> | MASF        | 26.9 | 56.8 | 68.2 | 78.3 | 92.0 | 65.4 | 64.6 |
> | MLDG        | 26.7 | 56.6 | 68.9 | 80.2 | 88.2 | 63.2 | 64.0 |
> | CCSA        | 27.6 | 57.4 | 68.3 | 75.0 | 90.5 | 67.6 | 64.4 |
> | LDDG [2]      | 27.9 | 60.1 | 69.7 | 81.7 | 92.7 | 69.8 | 67.0 |
> | REMA (Ours) | 29.1 | 63.4 | 70.8 | 83.5 | 94.8 | 75.3 | 69.5 |
>
> **References**
>
> [1] Domain Generalization using Causal Matching. In ICML, 2021.
>
> [2] Domain Generalization for Medical Imaging Classification with Linear-Dependency Regularization. In NeurIPS, 2020.
>
> [3] Mix and Reason: Reasoning over Semantic Topology with Data Mixing for Domain Generalization. In NeurIPS, 2022.

---

> > ### Comment · Reviewer_rhAB · 2024-08-11
> > **Thanks for the rebuttal**
> >
> > After reviewing the rebuttal and considering the comments from other reviewers, I will raise my score. My questions have been satisfactorily addressed. Thank you to the authors.

---

> > > ### Author Response · Authors · 2024-08-13
> > >
> > > We are glad to hear that our response helped resolve your questions. Thank you again for your time and constructive feedback!

---

### Official Review · Reviewer_M7aK · 2024-07-13

**Soundness:** 3
**Presentation:** 3
**Contribution:** 3
**Rating:** 6
**Confidence:** 3

**Summary:**

The authors propose an approach to tackle OOD generalization via a method that tightly combines learning-based feature extraction with graph-based relationship modelling to explicitly learn and represent the topological structure of image data, achieving competitive results across a range of OOD and test-time adaptation benchmarks.

**Strengths:**

**Originality & Significance:**
- Interesting method focusing on explicitly modelling the often neglected /only implicitly modelled topological structure of image data via a mixture of learning- and graph-based methods
- Proposed approach demonstrates an appealing way of ‘casting’ the common human intuition regarding our visual reasoning process, which draws on different levels of hierarchy, into an end-to-end trainable algorithm through a well-justified composition of different components

**Quality:**
- Experiments conducted with a good selection of comparative methods to gauge performance improvements across two different tasks (OOD and test-time adaptation) and multiple datasets
- The authors perform an appropriate ablation of their main components, both quantitatively and qualitatively

**Clarity:**
- The paper is well written and easy to read and follow; The topic is well motivated and well-placed within the context of related efforts, clearly pointing out which areas of research the authors build upon and what the remaining challenge is;
- Clear visualizations help the reader to quickly grasp the underlying concepts, both methodologically and architecturally

**Weaknesses:**

_TLDR; I do not see any severe ‘prohibitive’ weaknesses in this work, but have a few questions and requests that I’d like the authors to clarify & address._

- Some insight into what the slots actually represent would be insightful and elevate the paper, see below.
- Missing detail on complexity, please see question section below.
- Missing details regarding inference and training procedure, see below.
- Some inconsistencies in notation as well as some typos that should be corrected

**Questions:**

**Main concerns, questions & potential improvements:**

**[Q1]**: Consistency in notation could be improved.
 The authors initially use lower-case characters to indicate functions, e.g. in equation (1), and mathcal font to indicate sets (e.g. input, output, feature space/domain);
However, this suddenly changes during the introduction of the method (l.94),  and mathcal is used to refer to transformations (e.g. K_beta, which is nothing else than a function);
Then slightly later (l.135), mathcal refers to the ‘result’ of the linear transformation that essentially forms the graph nodes;
$\textrightarrow$ I’d highly recommend to keep a consistent notation to avoid confusion (as it did confuse me);

**[Q2]**: Some missing details regarding training & inference procedure.
- How exactly is training performed? The authors mention that the “algorithm first trains deep models using the reconstruction objective” (l.171) – is the model (or a part of it) then frozen when the graph is created/HORR trained, or is HORR simply added and everything then trained/finetuned? The appendix section doesn’t make this clear to me either, and some more details here would be helpful.
- The authors mention in l.99 that the queries “will be refined during the T attention iterations”. How many iterations are employed in practice, and is this performed for each new image pair (i.e. each step) during training?
- How exactly is the inference actually performed at test time? Does inference require pairs as employed during training, or what is the exact setup there?

**[Q3]**: Details around complexity;
Following up from Q2, what is the ‘complexity’ of the method in contrast to other related methods, e.g. in terms of inference time?
Note that given the impressive performance and graph-based nature, this could be an interesting insight to the reader (to gauge potential trade-offs between multi- and 1-step reasoning methods)

**[Q4]**: Details regarding slots / information bottleneck:
- The authors mention in the appendix that ‘5’ slots have been typically used. Is this the same for all datasets? And how many were selected as relevant?
- Do these numbers change across datasets or even classes? And if so, could you provide possible insights into why, e.g. multiple objects / increased complexity / more components, or similar.

**[Q5]**: Insights into the actual 'slot correspondence'/representation:
- 5 slots seems quite few to represent the content of an image. Is this due to the simplicity of the images, and do the authors have some intuition how this would change for `real-world' natural settings?
- How do the slots actually 'align' with 'components' of objects: Taking the motivational picture of the horse, would your algorithm actually identify 'parts' of an object as a component, or rather different objects in an image, or entirely different? Some insight into these aspects would be highly interesting to actually see to which extent the inner workings align with what we would expect from humans (which is, after all, your underlying motivation)

---

**Additional comments:**
- typo l.92: embedding -> embeddings
- capitalization l.109: we -> We
- typo l.216: methods […] is -> are

**Limitations:**

Limitations have been adequately addressed by the authors in the appendix;
$\textrightarrow$ I highly appreciate the authors being honest and providing ‘proper’ limitations to their method in terms of potential applicability!

---

> ### Author Rebuttal · Authors · 2024-08-06
>
> We are grateful for your insightful comments and appreciation of our work. We address each question in detail and provide further clarifications below.
>
> > **Q1: Consistency in notation.**
>
> Sorry for the confusion. We have modified the notation in Eq. (2) to make it consistent with other formulas, i.e., Q_gamma -> f_q, K_beta -> f_k, and V_phi -> f_v.
>
> > **Q2-1: How exactly is training performed?**
>
> We employ a phased training strategy, starting with training SSR to enable the model to accurately extract sparse representations from images, followed by fine-tuning with HORR to equip the model with the capability to reason about these main components. The model (or a part of it) is not frozen during HORR training.
>
> > **Q2-2: How many iterations are employed in practice, and is this performed for each new image pair (i.e. each step) during training?**
>
> The number of iterations is fixed at 3, consistent with the original slot attention mechanism. This process aims to progressively reconstruct the original image. In the context of slot attention using iterative attention, the process is generally applied repeatedly at each step during training for every new image pair.
>
> > **Q2-3: How exactly is the inference actually performed at test time? Does inference require pairs as employed during training, or what is the exact setup there?**
>
> During the inference phase, pairwise samples are not required; a single image or a batch of randomly sampled images suffices. For OOD generalization, a vanilla inference process is performed. For test-time adaptation, the model parameters need to be updated online using the test samples.
>
> > **Q3: Details around complexity.**
>
> We have included quantitative results regarding the complexity of the method in **the attached PDF (Fig. 4 and Fig. 5)**. When the iterative attention and graph matching are only performed during training, the inference speed (without TTA) is nearly equivalent to ERM. With TTA **( Fig. 5)**, our method's inference speed is also comparable to several mainstream approaches.
>
> > **Q4: Details regarding slots / information bottleneck**.
>
> As stated in the appendix, the initial number of slots is set to 5. In our experiments, this is the same for all datasets since OOD generalization and TTA benchmarks typically consist of single object images. Thus, there is no need to increase its value which will bring more computational overhead. However, for scene images, such as semantic segmentation datasets, we need more slots to present their composition. In a nutshell, the number of slots depends on the complexity of data and is relatively insensitive to the change of datasets that have the same type.
>
> > **Q5: Insights into the actual 'slot correspondence'/representation.**
>
> For the first subquestion, **(1)** continuing from the previous discussion, the number of slots required is related not only to the complexity of the image content but also to the nature of the task itself. In our experiments, we focus on the relatively simple case of image classification tasks involving single objects, which require fewer slots. However, in scenarios containing multiple objects where tasks like object detection or semantic segmentation are desired, more than ten slots might be necessary, depending on the specific task. **(2)** It is important to note that more slots are not always better; on one hand, the computational cost increases, and on the other, there might be an over-segmentation issue in segmentation tasks (conversely, too few slots can lead to under-segmentation). However, for classification tasks, the granularity with which we recognize the main components of an object is flexible. For instance, in describing a person, we could (i) broadly identify the upper and lower body, (ii) recognize the head, torso, and limbs, or (iii) further divide specific parts such as the torso into more detailed segments. **(3)** In summary, classification problems require fewer slots and are less sensitive to the number of slots, whereas scene understanding tasks necessitate a larger and more sensitive allocation of slots.
>
> For the second subquestion, **(1)** our process of identifying main components is somewhat akin to fully unsupervised attribute/concept discovery. That is, without component annotations, the model essentially learns an attention mask, where each position's value reflects its significance in relation to the class label. However, we cannot guarantee that the learned components will perfectly align with concepts readily understood by humans, as we lack fine-grained human annotations. Of course, replacing the encoder with a more powerful visual extractor like DINOv2 [1] could enhance discovery capabilities, but this would change both the whole experiment and the baseline methods. **(2)** From a methodological perspective, our proposed method can decompose images into high-level concepts in an unsupervised manner and cluster the images based on those discovered concepts. In **the attached PDF (Fig. 1)**, we provide some visual results from real-world datasets, showing that REMA can segment images into different areas (the number of areas depends on the number of slots). These areas are informative and correspond to different high-level concepts. For example, distinguishing between an animal's head, body, and legs. **(3)** As seen in Fig. 4(a) of the main paper, without REMA, the model might learn only small discriminative areas or even background regions. However, REMA enables the learning of objectness, which more completely emphasizes the entire foreground object area. In addition, the affinity matrix in Fig. 5 demonstrates that our REMA is capable of learning more accurate cross-domain correspondences, illustrating its robustness to distribution shifts.
>
> **Reference**
>
> [1] DINOv2: Learning Robust Visual Features without Supervision. In arXiv:2304.07193.

---

> > ### Comment · Reviewer_M7aK · 2024-08-12
> > **Thank you for the responses.**
> >
> > I'd like to thank the authors for their responses and the additional provided information & insights -- especially the visualisations regarding slots & image regions;
> >
> > Having read the other reviews and rebuttal, I will stick with my original rating and recommend weak acceptance

---

> > > ### Author Response · Authors · 2024-08-13
> > >
> > > Thank you for your insightful feedback and constructive comments, which have been invaluable in enhancing our manuscript. We will incorporate the additional results and discussions in the final version.

---

### Official Review · Reviewer_jWEa · 2024-07-13

**Soundness:** 4
**Presentation:** 2
**Contribution:** 4
**Rating:** 7
**Confidence:** 2

**Summary:**

A new methodology, REconstruct and MAtch (REMA) is introduced to learn a more robust and generalizable feature set in computer vision models.  REMA relies on a slot attention module to learn sparse embeddings of features which characterize a target object, and this module is then coupled with a high-order relational reasoning (HORR) module which creates a graph representation of the object.  This graph representation encodes how the sparse features relate to each other.  Because the feature encodings are both simplified through the sparse attention module and contain a learned topological relation from the HORR module, they are expected to be robust to domain drift. The robustness to domain drift is demonstrated using several benchmark datasets with consistent networks and training hyperparameters shared between them.  REMA consistently and clearly outperforms SOTA methods for OOD generalization; the results are convincing.

**Strengths:**

The methodology is novel; the combination of the two modules represents a new contribution to the area of identifying robust features for computer vision.  The results reported are extensive, including latent space analysis of how features are represented, ablation studies, and comparison against $\approx$20 other OOD generalization methods, with the proposed REMA method outperforming all of them.  An improved method for robust OOD model training is a significant contribution, and the work put into reporting the technical aspects of the methodology will encourage dedicated researchers to continue developing this approach to robust feature identification.

**Weaknesses:**

The report of the proposed method is exceedingly technical.  Discussion of the intuition motivating the method is restricted to a brief statement concerning the human visual system, and no connection back to the human visual system is made throughout the remaining discussion of the methodology.  No further motivation or context is provided for algorithmic design choices, which prevents the reader from understanding *why* the method works well. The emphasis is entirely on the *how*. I suspect that a significant literature review was performed as part of the algorithm design; sharing the literature review context for these design choices would have helped motivate and clarify the methodology.  It could easily have been included in the appendix without detracting attention from the main contribution of the work. Alternately, the context could have been included in the main result, and the implementation details left to the appendix.

The benefit of the HORR module is open to some question.  The impact of the HORR module vs the SSR module is discussed in Section 4.3, Table 4, and shown in Figure 6 with the tSNE embeddings.  The results of Table 4 are reported without uncertainty, and the values are close enough (within 2%-8.5% of the baseline without REMA) to make uncertainty a valuable indicator of the role of each module. Given the stochastic variation inherent to the tSNE algorithm, Figure 6c (SSR w/o HORR) and 6d (SSR w/ HORR) could be considered identical since there are no repeated embeddings reported, and no figure of merit describing the variance for repeated embeddings and clustering is included.  Visually, the embeddings are very close, so although the value of the SSR module is clearly demonstrated in disentangling latent features in Fig. 6c, the value of the HORR module has some doubt.  (An example of suitable metrics for quantifying the author's claim of better clustering in Figure 6d vs. Figure 6c might be DBSCAN or OPTICS applied to multiple embeddings, and quantified with the mean and std of the v-measure score for the clusters.)  The supplemental Figures 6-8 do little to clarify the situation; the authors state only that SSR and HORR behave differently depending on the type of corruption. An additional place to clarify the behavior of the individual SSR and HORR modules would have been in Figure 4a, where the grad cam results with and without the SSR and HORR modules could have been reported.  Finally, the comment in the supplement regarding training the SSR module first suggests that although including the HORR module may be beneficial, it is not necessarily key to the success of the work as no mention of training HORR first is made.

Overall, the results are extensive, and support the conclusion that REMA is an improvement on the SOTA for OOD robustness.  But the lack of contextualization and motivation for the algorithm in general, and the behavior of the different modules specifically, is a weakness of the paper.

**Questions:**

1. The stated goal of the Selective Slot-based Reconstruction (SSR) module is to create a sparse embedding of target features.  In section 3.3, it would seem that a variational auto encoder type approach is used to train the encoder.  Sparsity can already be considered forced due to the size of the feature vector/latent space of the VAE; a more sparse feature vector is just one that is smaller in this context.  How does the additional SSR module change the latent representations to make them more sparse?  Is it just removing specific frequency components?  Why is it not possible to do this directly on the latent space of the encoder through a specialized loss function without the use of an additional module?

2. In section 3.1 "(A) standard MLP skip connection is applied" after the GRU; why? What is the intuition here?

3. What is the minimum and maximum connectivity of the hypergraphs constructed in Section 3.2?  How would this dimensionality relate to the dimensionality of the sparse embeddings?  Of the original feature vector extracted from the encoding module?

4. Please provide citation(s) for the statement in Supplemental section C.1 "sparse modeling based on slots may struggle to accurately separate the scene into several main components".

5. In the supplemental section B.2, why is it reasonable to expect that the SSR and HORR modules have different effects based on the type of data corruption?  Shouldn't the keypoints of a target object (and therefore the sparse embedding of those key points as well as the topology of how those keypoints relate to each other) be unaffected by whether it is fog or frost corrupting the image?

**Limitations:**

The authors report limitations in the supplemental material.

---

> ### Author Rebuttal · Authors · 2024-08-06
>
> We are grateful for your insightful comments and appreciation of our work. We address each question in detail and provide further clarifications below.
>
> > **Q1:Motivation for algorithmic design choices (SSR + HORR).**
>
> (1) Although we aim to imitate the human vision process for OOD generalization, we cannot guarantee that the learned components will perfectly align with concepts readily understood by humans, due to the lack of fine-grained human annotations. To solve this issue, our SSR is a data-driven approach that enables the deep model itself to possess the capability of abstraction from input data. The model essentially learns an attention mask, where each position's value indicates its importance/association relative to the class label. Given the need to learn objectness or high-level concepts, slot attention naturally comes to mind as a classic solution for object-centric learning. By doing so, as shown in **the attached PDF (Fig. 1)**, the regions segmented by SSR mostly align with human understanding, such as distinguishing between an animal's head and body.
>
> (2) Having identified the main components, we naturally consider the relationships among them. Graph networks are a major tool for introducing relational inductive bias. Given that slots are sparse and require higher-order connections to fully capture the relationships, and since ordinary graphs can only model pairwise relationships, we opt for hypergraphs. Moreover, regarding how to associate objects of the same category across different domains, most previous methods utilize direct alignment (Fig. 1 of the main paper). However, now that we have identified the main components and their internal relationships, and both have been modeled as vertices and edges of a hypergraph, cross-domain connections are naturally achieved via graph matching.
>
> > **Q2: The benefit of the HORR module.**
>
> (1) Standard error in Tab. 4. In fact, the results in Table 4 are averaged over 3 random seeds. Please see **the attached PDF (Tab. 1)**.
>
> (2) w/o HORR vs. w/ HORR. As suggested, we have run t-SNE 10 times for each case and then applied DBSCAN to each embedding. Then, we calculate the mean and std of V-measure score: w/o HORR (Mean V-measure: 0.65, std of V-measure: 0.08) vs. w/ HORR (Mean V-measure: 0.79, std of V-measure: 0.03).
>
> (3) Additional Gram-CAM. We added these results to **the attached PDF (Fig. 2)**.
>
> (4) Training Sequence. HORR first will lead to an unstable training process due to the dense latent features (without using SSR). Thus, joint training or SSR first would be better choices for our work.
>
> > **Q3-1: A VAE-type approach is used to train the encoder.**
>
> As suggested, we trained an encoder using a VAE, using an equal number of mean and var as there are slots, for comparison. Please see **the attached PDF (Tab. 1)**. Noting that models with VAE achieve better performance than the ERM baseline, this validates our motivation to seek sparsity. However, there remains a performance gap compared to our SSR, demonstrating the superiority of our module.
>
> > **Q3-2: How does the additional SSR module change the latent representations to make them more sparse?**
>
> SSR actively reorganizes the latent space into discrete, interpretable slots, each capturing distinct and salient features of the input data. It structurally enhances sparsity by ensuring each slot is maximally informative and minimally redundant, thus facilitating sparse representations. This goes beyond just removing specific frequency components. Directly using a specialized loss function would lack this level of granularity and control. The modular nature of the SSR (data-driven) allows for targeted optimization and adaptation to diverse datasets.
>
> > **Q4: MLP skip connection.**
>
> This is a standard step in slot attention, where (1) MLP allows the model to learn complex patterns and relationships between the slots and the input data. (2) Skip connections aid in preserving the original information from the input throughout the network.
>
> > **Q5: Details about hypergraphs and the dimensionality of embedding.**
>
> As indicated in L143-144, the number of hyperedges is equal to the number of slots. The dimension of each slot is 256, matching the original feature dimension from the encoding module.
>
> > **Q6: Citation(s) for the statement in Supplemental section C.1**
>
> We aim to divide objects into several components based on class labels. However, for scenes without clear foreground and background distinctions, this approach encounters challenges. In cases of tuberculosis—a binary classification problem—the X-ray images exhibit diffuse characteristics w/o specific lesions like lung nodules, which are causally linked to conditions such as malignant nodules and lung cancer. Thus, the difference is related to the image's style, making it hard to segment the original image into distinct parts based on the presence or absence of disease. While we did not find literature specifically addressing this issue, there are some related studies (e.g., [1,2]) that can be referenced.
>
> [1] Unsupervised Learning of Discriminative Attributes and Visual Representations. In CVPR, 2016.
>
> [2] Bridging the Gap to Real-World Object-Centric Learning. In ICLR, 2023.
>
> > **Q7: Why is it reasonable to expect that the SSR and HORR modules have different effects based on the type of data corruption?**
>
> Sorry for the confusion. You are correct in pointing out that we aim to ensure that "the keypoints of a target object—and therefore the sparse embedding of those keypoints as well as the topology of how those keypoints relate to each other—remain unaffected by external factors such as fog or frost," emphasizing topological homogeneity. We will revise this paragraph for clarity.

---

### Author Rebuttal · Authors · 2024-08-06

We sincerely appreciate all four reviewers for their time and effort in providing feedback and suggestions on our work. We are glad that reviewers recognize our paper to be *novel* (jWEa, M7aK), *well-motivated* (M7aK, rhAB, EaaQ), and performing *extensive experiments and ablation studies* (jWEa, M7aK, rhAB).

We have addressed the comments and questions in individual responses to each reviewer. The main changes we made include:

- We have provided additional visual results and experimental comparisons, explained the design motivations of our algorithm, and offered insights into the proposed SSR and HORR (jWEa).
- We clarified some design details, discussed complexity, and provided insights into the actual 'slot correspondence' and representation (M7aK).
- We conducted both quantitative and qualitative discussions on parameter sensitivity, computational efficiency, and performance in real-world scenarios (rhAB).
- We provided clarifications to eliminate significant misunderstandings about the problem definition, baseline selection, and experimental comparisons (EaaQ).

If you have any further questions or require additional clarification, please feel free to raise them during the author-reviewer discussion phase. Thank you!

---

### Decision · Program_Chairs · 2024-09-25

**Decision:**

Accept (spotlight)

**Comment:**

This paper received an Accept, two Weak Accepts, and a Borderline Accept. All reviewers agree that the work presents an interesting and innovative approach. Notably, they commend the novelty and well-motivated design of the selective slot-based reconstruction module and the hypergraph-based relational reasoning framework. After the rebuttal, the authors successfully addressed all the concerns and questions raised by the reviewers, resulting in all reviewers being satisfied post-rebuttal. Consequently, the AC has decided to Accept this paper, recognizing its potential impact on both the theoretical and practical aspects of out-of-distribution generalization tasks.